# A Near-Optimal Energy Management Mechanism Considering QoS and Fairness Requirements in Tree Structure Wireless Sensor Networks

**DOI:** 10.3390/s23020763

**Published:** 2023-01-09

**Authors:** Kuang-Yen Tai, Bo-Chen Liu, Chiu-Han Hsiao, Ming-Chi Tsai, Frank Yeong-Sung Lin

**Affiliations:** 1Department of Information Management, National Taiwan University, Taibei 106, Taiwan; 2Research Center for Information Technology Innovation, Academia Sinica, Taibei 115, Taiwan

**Keywords:** wireless sensor network, QoS, sensor activity, Lagrangian Relaxation, energy consumption

## Abstract

The rapid development of AIOT-related technologies has revolutionized various industries. The advantage of such real-time sensing, low costs, small sizes, and easy deployment makes extensive use of wireless sensor networks in various fields. However, due to the wireless transmission of data, and limited built-in power supply, controlling energy consumption and making the application of the sensor network more efficient is still an urgent problem to be solved in practice. In this study, we construct this problem as a tree structure wireless sensor network mathematical model, which mainly considers the QoS and fairness requirements. This study determines the probability of sensor activity, transmission distance, and transmission of the packet size, and thereby minimizes energy consumption. The Lagrangian Relaxation method is used to find the optimal solution with the lowest energy consumption while maintaining the network’s transmission efficiency. The experimental results confirm that the decision-making speed and energy consumption can be effectively improved.

## 1. Introduction

### 1.1. Background

In this study, we designed a tree structure mathematical model for wireless sensor networks (WSNs) that reduces delays in data transmission and aggregation to optimize the energy efficiency of WSNs. With the rapid development of microelectromechanical systems and wireless network communication technology, the application of WSNs has increased [1]. WSN is a network system composed of multiple sensor nodes and is used, for example, in environmental detection, military, smart home, and health-care applications. For mass deployment, wireless sensing nodes must be small and inexpensive, and have low power consumption [2]. The energy limitation of each sensor node and the energy load balance of the network nodes directly affects the lifespan of the entire WSN. One factor influencing the applicability of a WSN includes the additional energy consumption caused by the aggregation and retransmission of data. Load imbalance is caused by a heavy load on an individual sensor, resulting in greater energy consumption through transmission. Due to the energy limitations of each sensor node directly affecting the lifespan of the entire WSN, reducing energy consumption is critical to prolong the network’s lifespan [3].

In practical applications, sensor nodes have different tasks due to the differences in the sensor nodes’ environment. A heterogeneous sensor network further leads to different energy loads. To increase network lifespan, reducing energy consumption is crucial [4]. Data aggregation involves transmitting sensing data to specific aggregation nodes. It reduces the number of times each sensing node transmits to the base station, yielding energy savings [5].

Figure 1 shows the general architecture of a WSN with sensor nodes in coverage areas. The sensor nodes generate data and transmits it to the relay nodes. The relay nodes will do preliminary processing that discriminates the correctness and usefulness of the data and further transmit it to the sink nodes. The sink node will aggregate data from relay nodes and do more advanced processing, such as classification. Finally, the data that needs to be applied or stored will be sent to the cloud data center. Each node communicates with higher-level nodes to transmit collected data through network interfaces. The location of higher-level nodes is determined according to the range of the sink or relay nodes, which gathers data from lower-level nodes. Network data are usually processed by the sink nodes or relay nodes, which only sends data that is relevant to the user. High-level nodes are located closer to the cloud data center and consume more energy than lower-level nodes due to their high usage for data aggregation [6]. Thus, sensor nodes optimize several variables, including the amount of data flow, the transmission power level, and the activation of each sensor to minimize energy consumption.

### 1.2. Research Motivation

The sink node in the tree structure sensor network can be static, dynamically assigned, or even with mobile capabilities, e.g., associated with a robot or a drone. The motivation of this study is to propose a mathematical model when the channel quality between a top-layer sensor node and the designated sink node is known. The proposed approach then can efficiently and effectively calculate a near-optimal power control policy for each sensor node in the network subject to fairness and end-to-end QoS constraints to preserve power consumption well and prolong the lifetime of the entire system. The proposed approach will apply to various application environments and sink node types. In the point of dynamic WSN, some related studies have begun to focus on mobile sink nodes in recent years [6,7] and find optimal roving paths for robot sink nodes [8]. It solves the limited energy problem of traditional sensors. However, there are still many practical difficulties of charging failure or energy limitations in making mobile robot sink nodes move between nodes to provide power in these extreme environments. Therefore, there will be suitable solutions in different application scenarios. On the other hand, in some extreme environments or where the robot cannot move freely, placing the sensors with solid environmental tolerance in a static location is necessary. A tree structure will be formed at a specific time in static or dynamic WSNs and extend the concept of the tree structure to establish an energy optimization algorithm that can be adjusted according to the current situation.

Different application scenarios or environmental changes might make the topology of the WSN unbalanced, causing the lifetime of the nodes and the Qos to be reduced [9]. WSNs are essential in developing many applications for AI and the Internet of Things (AIoT). Wireless sensors are usually low-power, and their power supply is usually not supplementary, so if the energy consumption is exhausted, there will be loopholes in the original complete detection network [10,11].

### 1.3. Main Contributions

This study developed a mathematical model with a practical application point of view and investigated the activation states of nodes and the trade-off between energy consumption and connectivity. The efficiency of retransmission and acknowledgment protocols on the data link layer was also considered. A tree structure was used to form a complete WSN that accounts for both packet size and transmission power. The results show that the decision-making speed and energy consumption can be effectively improved.

### 1.4. Organization of the Paper

The organization of this paper is described as follows. After the introduction in Section 1 above, Section 2 introduced the research topics and practices in current research. The mathematical definition of the proposed model considering QoS and fairness requirements will be described in Section 3. Section 4 will provide the optimal solution approach for the problem stated in this study. Experimental results and a discussion of the previously proposed methods will be shown in Section 5. Finally, the conclusion and future work of this study will be discussed in Section 6.

## 2. Related Works

Mechanisms to reduce energy consumption in WSNs have been studied extensively and most have centered on the scheduling of sensors awareness or data transmission. Power control mechanisms optimize the trade-off between energy consumption and connectivity by alternating between sleep and active node states can reduce energy waste [12,13,14]. Finding optimal routes, clusters, and aggregation points when designing sensor networks also prevents energy waste. Some commonly used methods are discussed in this section.

### 2.1. Duty Cycle

A duty cycle manages the energy consumption of the nodes by constantly switching the states of the sensor nodes. The sensor node has different operation modes, such as idle, sleep, listen, and transmit. Distance-based duty cycle assignment is based on the assumption that nodes closer to the sink must transmit a larger packet than those farther from the sink and is used to determine the duty cycle on the basis of the distance of a given node from the sink node [15]. Traffic-adaptive distance-based duty cycle assignment (TDDCA) is an improvement on traditional duty cycle–based methods and is based on the traffic patterns observed by the nodes [16,17]. Receiver-based protocols indicate network traffic on the basis of the number of retransmitted packets. If retransmission increases dramatically and the original packet number is exceeded, traffic congestion may occur. TDDCA then tunes the duty cycle to mitigate congestion [18].

Coverage requirements are also related to the node duty cycle. Ideally, the two goals of minimizing energy consumption and ensuring coverage of the sensor nodes over the target areas should be satisfied. A disjoint set of nodes that covers the monitored area maximizes network lifespan [19,20].

### 2.2. Transmission Power Control

The control of transmission power methods involves adjusting the sensor node’s transmission power to appropriate levels according to the range between the transmitter and receiver or traffic status [21]. The protocol layers employed are classified by their respective approach to power control into media access control (MAC), network, and transport layers [22,23]. A MAC layer approach is designed to reduce the chance of collision to minimize energy consumption in transmission. The network layer approach is based on two technologies: power-aware routing and maximum lifespan routing [24,25]. Transport layer protocols are used to control congestion and retransmission in a network to reduce energy consumption. Power control methods are known as range assignment methods or strong minimum energy–topology methods, and they center on striking the best trade-off between throughput, traffic, and reliability [26].

In the adaptive transmission power control (ATPC) model, each node adjusts its power according to link quality [27,28]. This model employs a feedback-based transmission power control algorithm to maintain link quality dynamically [29]. ATPC depends on pairwise adjustment to provide energy savings, and this demonstrates the superiority of the link level compared with the node network levels in terms of delivering greater energy efficiency [30].

### 2.3. Topology Control and Routing in Sensor Networks

Topology control can reduce network traffic, avoid packet collision, improve network throughput, and save energy. Central to node deployment is finding a subset of strongly connected nodes to serve as the network’s backbone [31]. The remaining nodes connect to the backbone. This topology of the backbone guarantees the network’s connectivity; this topology allows non-backbone nodes to be turned off to save energy and is often mathematically modeled as a connected dominating set (CDS) problem [32,33]. A CDS-based algorithm is used to construct a network, prolong its lifespan, and balance energy consumption [34]. 

Routing (data transmission) has been investigated in a broad range of studies. It can be roughly divided into shortest path tree (SPT)-based models and minimum spanning tree (MST)-based models; these models center on addressing flow problems [34,35,36,37]. SPT and MST algorithms are used to find paths that yield improvements in energy efficiency. The characteristics of each node are considered, such as their energy consumption [38]. However, SPTs may lead to an imbalanced load among sensors. Routing problems that minimize total energy consumption, or maximize network lifespan, are also formulated as multicommodity flow problems [39,40]. The commodity is a source–destination pair, indicating that the multicommodity flow problem is NP-hard. Multicommodity flow problems are formulated using integer linear programming and represent flow using the number of packets and transmission energy [41,42,43,44,45].

### 2.4. The Proposed Approach

This paper’s innovative viewpoint on the balance of transmission between sensors at different levels is its fairness to achieve energy consumption management that is closer to the practical field [46]. The research mentioned above on WSN provides many relevant issues worthy of reference.

A WSN was usually designed for a particular application with a specific topology [47]. The structure of the network needs to be adjusted according to different application scenarios, and the most common one in practical applications is the tree structure. However, with different application scenarios or environmental changes, when there are sensors in the structure with unbalanced loads, the lifetime of the nodes will be reduced, and the energy consumption of the overall network will be significantly increased [48,49,50,51]. Unequal communication and data processing distribution among sensors results in energy consumption balancing issues, especially in network structures with multi-stage communication [52]. Often touched, sink or relay sensor nodes require more energy consumption than other nodes that perform additional data processing tasks. To simultaneously consider the fairness of loading nodes at different levels in the network and the aggregation and acquisition of data retention is a challenging problem that has seldom been considered in previous studies [53].

## 3. Mathematical Model

This section introduces the tree structure WSN model proposed in this study using mathematical programming, including problem definition and the system model with constraints. Specifically, to formulate network structures with multi-stage communication and consider transmission balance and QoS.

### 3.1. Problem Definition

This study formulated a mathematical model where the aim is to minimize the energy consumption of a WSN by determining the optimal connectivity and throughput. The model was designed as a time-slotted system that finds the optimal solution based on the relationship between all nodes in a tree structure sensor network. In a time-slotted system, a collision occurs if more than two node pairs compete for the same slot, and each time slot’s bandwidth is considered to be fixed. The parameters and decision variables used are shown in Table 1 and Table 2:

The functions $C_{\theta_{i}}^{a}(r_{\theta_{i}},m)$, $C_{\theta_{i}}^{b}(r_{\theta_{i}},m)$, $C_{\kappa_{i}}^{a}(r_{\kappa_{i}},m)$, $C_{\kappa_{i}}^{b}(r_{\kappa_{i}},m)$, $C_{\zeta }^{a}(r_{\zeta },m)$, $C_{\zeta }^{b}(r_{\zeta },m)$, $C_{\theta_{i}}^{\tau }(r_{\theta_{i}},m)$ and $C_{\kappa_{i}}^{\tau }(r_{\kappa_{i}},m)$ are defined as variables, despite having assigned values.

### 3.2. System Model

The proposed WSN tree structure model is shown in Figure 2. This study used a network tree structure with sensor nodes *θ*, relay nodes *κ*, and sink nodes *ζ*. The network tree structure is a data-centric network containing *n* subtrees and one sink node; each subtree contains *v* sensor nodes and one relay node.

The sensor nodes have varying levels of importance and priority due to the different areas they monitor. This model aims to aggregate data from all sensors into the sink in a manner that minimizes energy consumption and satisfies the average delay constraints in all origin–destination pairs. This delay is the maximum link delay of all possible links. Sensor nodes send data to relay nodes. This model rests on the assumption that sensor nodes send data as soon as they are activated. Relay nodes aggregate sensor node data and transmit it to the sink when they are active. The only purpose of sink nodes in this model is to receive data. We assume that each node is parallel and competing for an interface in a time slot. Every sensor has information to send and sends it whenever it is able to.

The proposed model’s objective function (1) determines the packet size transmitted by node *i*. It minimizes the power consumption under the constraints of considering the trade-off between packet size and power consumption for a single byte of the overall tree structure WSN. The overall expression is rendered as a convex function as a larger packet size results in a larger time slot. The physical meaning in the function represents the probability of transmitting a packet without error decreases as the possible packet size, *m*, increases, under the assumption that each packet has a fixed-size header—a larger packet size results in a larger throughput. The objective function is divided by *m* for normalization in terms of the time slot length.
(1)$$\min\sum_{i\in N}\left[\begin{array}{l} \frac{C_{\theta_{i}}^{a}(r_{\theta_{i}},m)q_{\theta_{i}}v+C_{\theta_{i}}^{b}(r_{\theta_{i}},m)(1-q_{\theta_{i}})v+C_{\theta_{i}}^{\tau }(r_{\theta_{i}},m)q_{\theta_{i}}v}{m}\\ +\frac{C_{\kappa_{i}}^{a}(r_{\kappa_{i}},m)q_{kR_{i}}+C_{\kappa_{i}}^{b}(r_{\kappa_{i}},m)(1-q_{kR_{i}})+C_{\kappa_{i}}^{\tau }(r_{\kappa_{i}},m)q_{kS_{i}}}{m}\\ +\frac{C_{\zeta }^{a}(r_{\zeta },m)q_{\zeta }+C_{\zeta }^{b}(r_{\zeta },m)(1-q_{\zeta })}{m} \end{array}\right]$$

This study added the following constraints to describe the problem more precisely:

Equation (2) was developed to ensure in the QoS of the network, that the time taken for a single successful transmission from the sensor nodes to sink node *ζ* is shorter than the allowable delay. Successful transmission is assumed as the process of transmitting from a sensor node (*n*) to a sink node (*ζ*), via a relay node (*v*). This is calculated as the number of transmissions before the first successful transmission multiplied by the timeout interval for each *τ_θ__i_* plus one time slot for the successful transmission. We also added the transmission time from the relay node to the sink node, which is multiplied by the timeout interval for each transmission *τ_κ__i_* plus one time slot for the successful transmission. The overall transmission time should be lower or equal to a given allowable end to end delay from sensor node to sink node, it us assure the service quality of the overall network in different application scenarios.
(2)$$\tau_{\theta_{i}}\left[\frac{1}{q_{\theta_{i}}{\left(1-q_{\theta_{i}}\right)}^{\left(v-1\right)}q_{\kappa R_{i}}P_{\theta_{i},\kappa_{i}}(r_{\theta_{i}},m)}-1\right]+\tau_{\kappa_{i}}\left[\frac{1}{q_{\kappa S_{i}}{\left(1-q_{\kappa S_{i}}\right)}^{\left(n-1\right)}q_{\zeta }P_{\kappa_{i},\zeta }(r_{\kappa_{i}},m)}-1\right]+2\leq \bar{T_{\theta_{i},\zeta }}\;\forall i\in N,m\in M$$

The input and output throughput were calculated in Equations (3) and (4), and they were combined in Equation (5) to guarantee the fairness of the tree structure sensor network by limiting the probability of being active in each sensor nodes. The tree structure sensor network contains relay nodes responsible for transferring data collected from the sensor nodes. Hence, the throughput constraint will ensure packets are not discarded. We assume that relay nodes have a certain amount of capacity to store those data that were unable to be sent temporarily and will always send when there is a chance, also to ensure the fairness requirement that make sure any node is not overused. Thus, the average output throughput should be greater or equal to the average output throughput to avoid buffer overflow.

The input throughput of the relay node is composed of the summation output of all sensor nodes. The size of the packet sent by sensor nodes is multiplied by the number of time slots needed for a single successful transmission to pass the data to the relay node times the number of sensor nodes, shown as Equation (3):(3)$$m\times v\times q_{\theta_{i}}{\left(1-q_{\theta_{i}}\right)}^{\left(v-1\right)}q_{\kappa R_{i}}P_{\theta_{i,}\kappa_{i}}(r_{\theta_{i}},m)$$

The output throughput of the relay node is the size of the packet sent by the relay node is multiplied by the number of time slots needed for a single successful transmission to pass the data to the sink node, shown as Equation (4):(4)$$m\times q_{\kappa S_{i}}{\left(1-q_{\kappa S_{i}}\right)}^{\left(n-1\right)}q_{\zeta }P_{\kappa_{i},\zeta }(r_{\kappa_{i}},m)$$

Equation (5) ensures that the output throughput of the relay node is greater than the input throughput to prevent overflowing.
(5)$$mq_{\kappa S_{i}}{\left(1-q_{\kappa S_{i}}\right)}^{\left(n-1\right)}q_{\zeta }P_{\kappa_{i},\zeta }(r_{\kappa_{i}},m)\geq mvq_{\theta_{i}}{\left(1-q_{\theta_{i}}\right)}^{\left(v-1\right)}q_{\kappa R_{i}}P_{\theta_{i,}\kappa_{i}}(r_{\theta_{i}},m)\;\forall i\in N,m\in M$$
where $q_{\theta_{i}}$, $q_{\kappa R_{i}}$, $q_{\kappa S_{i}}$, $q_{\zeta }$, $P_{\theta_{i},\kappa_{i}}(r_{\theta_{i}},m)$ and $P_{\kappa_{i},\zeta }(r_{\kappa_{i}},m)$ denote the probability that node *θ_i_* is active in a given time slot, the probability that node *κ_i_* is active for receiving data in a given time slot, the probability that node *κ_i_* is active for transmission in a given time slot, the probability that node *ξ* is active in a given time slot, the probability of node *θ_i_* transmitting the packet to node *κ_i_* without error with a transmission range radius of *r_θi_*, and the probability of node *κ_i_* transmitting the packet to node *ξ* without error with a transmission range radius of *r_κi_*, respectively. The values of $q_{\theta_{i}}$, $q_{\kappa R_{i}}$, $q_{\kappa S_{i}}$, $q_{\zeta }$, $P_{\theta_{i},\kappa_{i}}(r_{\theta_{i}},m)$ and $P_{\kappa_{i},\zeta }(r_{\kappa_{i}},m)$ range between *ϵ* and 1.
(6)$$\varepsilon \leq q_{\theta_{i}}\leq 1\;\;\;\;\;\;\forall i\in N$$
(7)$$\varepsilon \leq q_{\kappa R_{i}}\leq 1\;\;\;\;\;\;\forall i\in N$$
(8)$$\varepsilon \leq q_{\kappa S_{i}}\leq 1\;\;\;\;\;\;\forall i\in N$$
(9)$$\varepsilon \leq q_{\zeta }\leq 1$$
(10)$$\varepsilon \leq P_{\theta_{i},\kappa_{i}}(r_{\theta_{i}},m)\leq 1\;\;\;\;\forall i\in N,m\in M$$
(11)$$\varepsilon \leq P_{\kappa_{i},\zeta }(r_{\kappa_{i}},m)\leq 1\;\;\;\;\forall i\in N,m\in M$$

To account for the given transmission range of each sensor node, Equations (12)–(14) were defined as follows.
(12)$$r_{\theta_{i}}\in R_{\theta_{i}}\;\;\;\;\;\;\forall i\in N$$
(13)$$r_{\kappa_{i}}\in R_{\kappa_{i}}\;\;\;\;\;\;\forall i\in N$$
(14)$$r_{\zeta }\in R_{\zeta }$$

This study assumes that when a single node loses power, the WSN is considered dysfunctional and the system’s lifespan goal is thus achieved. A conditional limit was included based on the initial power of each sensor node to ensure that every node reaches the system’s target lifespan. We defined Equations (15)–(17) as follows to ensure that the lifespan of the nodes exceeds the expected lifespan of the sensor network.
(15)$$[C_{\theta_{i}}^{a}(r_{\theta_{i}},m)q_{\theta_{i}}+C_{\theta_{i}}^{b}(r_{\theta_{i}},m)(1-q_{\theta_{i}})+C_{\theta_{i}}^{\tau }(r_{\theta_{i}},m)q_{\theta_{i}}]t\leq P_{\theta_{i}}\;\;\forall i\in N$$
(16)$$[C_{\kappa_{i}}^{a}(r_{\kappa_{i}},m)q_{kR_{i}}+C_{\kappa_{i}}^{b}(r_{\kappa_{i}},m)(1-q_{kR_{i}})+C_{\kappa_{i}}^{\tau }(r_{\kappa_{i}},m)q_{kS_{i}}]t\leq P_{\kappa_{i}}\;\;\forall i\in N$$
(17)$$[C_{\zeta }^{a}(r_{\zeta },m)q_{\zeta }+C_{\zeta }^{b}(r_{\zeta },m)(1-q_{\zeta })]t\leq P_{\zeta }$$

## 4. Optimization Solution Approach

In this section, the optimal solution proposed in this study will be introduced within four subsections, including 4.1 the Lagrangian Relaxation method, the chosen optimization method, and Figure 3 shows the procedure flow chart of LR; 4.2 reformulation for LR optimization solution to reformulate the proposed mathematical model to make it so it can be decomposed into subproblems; 4.3 the solution of LR subproblems, which describes the solution approach of the LR subproblems; 4.4 obtaining the primal feasible solution, which introduces the proposed primal feasible solution and the process was shown in Figure 4.

The mathematical model proposed in this study considers various parameters, goals, and constraints. This model has high mathematical complexity, including nonlinear and integer conditional constraints, and conforms to the characteristics of NP-hard. The research adopts problem-solving skills to transform the original highly complex restriction formulas into decomposed mathematical models and sub-problems to apply high-performance and high-efficiency problem-solving methods, such as Lagrangian Relaxation.

### 4.1. Lagrangian Relaxation Method

This problem is mathematically intricate, and Lagrangian Relaxation (LR) was chosen as the approach for arriving at a solution in our optimization-based power control method. LR is a widely used tool in mathematical programming applications. The aim of LR is to relax constraints that are difficult to solve. By doing so, the original problem becomes an LR problem that is relatively easy to solve. We create this LR problem by moving difficult constraints to the primal objective function with their respective coefficients, namely Lagrangian multipliers. Lagrangian multipliers are used as penalties when constraints are violated [54]. The LR problem can be decomposed into several subproblems. This renders it easier to solve. Subproblems are decomposed from the LR problem by separating the constraints and parts of LR problems containing the same decision variables. Each subproblem is then solved optimally according to their characteristics [55]. The solution to the LR problem regarding decision variables is then obtained.

The solution obtained from the LR problem forms the lower bound (LB) in a minimization problem. Moreover, if the solution can be applied to the primal objective function, meaning that it does not violate any constraints, an upper bound (UB) is produced. The primal optimal solution lies between the LB and UB. If the solution to the LR problem is not feasible, the solution is tuned using heuristic methods to make it feasible. The gap between the LB and UB can also serve as an indicator of solution quality, where smaller gaps indicate higher solution quality, and the best solution is produced when the LB overlaps with the UB. Figure 3 shows the procedure of the LR method.

### 4.2. Reformulation for LR Optimization Solution

Decision variables were multiplied under serval constraints. This violated the convexity of the constraint, meaning the decision variables could not be separated. For an LR based solution, this study used a subsidiary decision variable and logarithmic function to solve the complex problem regarding the multiplication of decision variables as follows:(18)$$q_{\theta_{i}}{\left(1-q_{\theta_{i}}\right)}^{\left(v-1\right)}q_{\kappa R_{i}}P_{\theta_{i},\kappa_{i}}(r_{\theta_{i}},m)=Z_{\theta_{i},\kappa_{i}}$$
(19)$$q_{\kappa S_{i}}{\left(1-q_{\kappa S_{i}}\right)}^{\left(n-1\right)}q_{\zeta }P_{\kappa_{i},\zeta }(r_{\kappa_{i}},m)=Z_{\kappa_{i},\zeta }$$
(20)$$(1-q_{\theta_{i}})=I_{\theta_{i}}$$
(21)$$(1-q_{\kappa R_{i}})=I_{\kappa R_{i}}$$
(22)$$(1-q_{\kappa S_{i}})=I_{\kappa S_{i}}$$
(23)$$C_{\theta_{i}}^{a}(r_{\theta_{i}},m)\times q_{\theta_{i}}=x_{\theta_{i}}$$
(24)$$C_{\kappa_{i}}^{a}(r_{\kappa_{i}},m)\times q_{\kappa R_{i}}=x_{\kappa_{i}}$$
(25)$$C_{\zeta }^{a}(r_{\zeta },m)\times q_{\zeta }=x_{\zeta }$$
(26)$$C_{\theta_{i}}^{b}(r_{\theta_{i}},m)\times I_{\theta_{i}}=y_{\theta_{i}}$$
(27)$$C_{\kappa_{i}}^{b}(r_{\kappa_{i}},m)\times I_{\kappa R_{i}}=y_{\kappa_{i}}$$
(28)$$C_{\zeta }^{b}(r_{\zeta },m)\times q_{\zeta }=y_{\zeta }$$
(29)$$C_{\theta_{i}}^{\tau }(r_{\theta_{i}},m)q_{\theta_{i}}=\beta_{\theta_{i},\kappa_{i}}$$
(30)$$C_{\kappa_{i}}^{\tau }(r_{\kappa_{i}},m)q_{\kappa S_{i}}=\beta_{\kappa_{i},\zeta }$$
(31)$$Z_{\theta_{i},\kappa_{i}}\times Z_{\kappa_{i},\zeta }=D_{\theta_{i},\zeta }$$

After reformulation, the constraints that needed to be solved were relaxed using the LR problem with a logarithm function and respective Lagrangian multipliers (*μ*). The LR problem is presented as follows.
$$\begin{array}{l} \min\sum_{i\in N}\left(\frac{x_{\theta_{i}}\times v+y_{\theta_{i}}\times v+\beta_{\theta_{i},\kappa_{i}}\times v+x_{\kappa_{i}}+y_{\kappa_{i}}+\beta_{\kappa_{i},\zeta }+x_{\zeta }+C_{\zeta }^{b}(r_{\zeta },m)-y_{\zeta }}{m}\right)\\ +\sum_{i\in N}\mu_{i}^{1}(\log q_{\theta_{i}}+(v-1)\log I_{\theta_{i}}+\log_{q_{\kappa }R_{i}}+\log P_{\theta_{i},\kappa_{i}}(r_{\theta_{i}},m)-\log Z_{\theta_{i},\kappa_{i}})\\ +\sum_{i\in N}\mu_{i}^{2}(\log q_{\kappa S_{i}}+(n-1)\log I_{\kappa S_{i}}+\log q_{\zeta }+\log P_{\kappa_{i},\zeta }(r_{\kappa_{i}},m)-\log Z_{\kappa_{i},\zeta })\\ +\sum_{i\in N}\mu_{i}^{3}(\log C_{\theta_{i}}^{a}(r_{\theta_{i}},m)+\log q_{\theta_{i}}-\log x_{\theta_{i}})+\sum_{i\in N}\mu_{i}^{4}(\log C_{\kappa_{i}}^{a}(r_{\kappa_{i}},m)+\log q_{\kappa R_{i}}-\log x_{\kappa_{i}})\\ +\sum_{i\in N}\mu^{5}(\log C_{\zeta }^{a}(r_{\zeta },m)+\log q_{\zeta }-\log x_{\zeta })+\sum_{i\in N}\mu_{i}^{6}(\log C_{\theta_{i}}^{b}(r_{\theta_{i}},m)+\log I_{\theta_{i}}-\log y_{\theta_{i}})\\ +\sum_{i\in N}\mu_{i}^{7}(\log C_{\kappa_{i}}^{b}(r_{\kappa_{i}},m)+\log I_{\kappa R_{i}}-\log y_{\kappa_{i}})+\sum_{i\in N}\mu_{i}^{8}(\log C_{\zeta }^{b}(r_{\zeta },m)+\log q_{\zeta }-\log y_{\zeta })\\ +\sum_{i\in N}\mu_{i}^{9}(I_{\theta_{i}}+q_{\theta_{i}}-1)+\sum_{i\in N}\mu_{i}^{10}(I_{\kappa R_{i}}+q_{\kappa R_{i}}-1)+\sum_{i\in N}\mu_{i}^{11}(I_{\kappa S_{i}}+q_{\kappa S_{i}}-1)\\ +\sum_{i\in N}\mu_{i}^{12}(\log C_{\theta_{i}}^{\tau }(r_{\theta_{i}},m)+\log q_{\theta_{i}}-\beta_{\theta_{i},\kappa_{i}})+\sum_{i\in N}\mu_{i}^{13}(\log C_{\kappa_{i}}^{\tau }(r_{\kappa_{i}},m)+\log q_{\kappa S_{i}}-\beta_{\kappa_{i},\zeta })\\ +\sum_{i\in N}\mu_{i}^{14}(\log Z_{\theta_{i},\kappa_{i}}+\log Z_{\kappa_{i},\zeta }-\log D_{\theta_{i},\zeta })\\ +\sum_{i\in N}\mu_{i}^{15}(\tau_{\theta_{i}}Z_{\kappa_{i},\zeta }+\tau_{\kappa_{i}}Z_{\theta_{i},\kappa_{i}}-\tau_{\theta_{i}}D_{\theta_{i},\zeta }-\tau_{\kappa_{i}}D_{\theta_{i},\zeta }+2D_{\theta_{i},\zeta }-D_{\theta_{i},\zeta }\bar{T_{\theta_{i},\zeta }})\\ +\sum_{i\in N}\mu_{i}^{16}(vZ_{\theta_{i},\kappa_{i}}-Z_{\kappa_{i},\zeta }) \end{array}$$(*LR*)

Subject to:(32)$$\varepsilon \leq q_{\theta_{i}},q_{\kappa R_{i}},q_{\kappa S_{i}},q_{\zeta }\leq 1\;\;\;\;\forall i\in N$$
(33)$$0\leq I_{\theta_{i}},I_{\kappa R_{i}},I_{\kappa S_{i}}\leq 1-\varepsilon \;\;\;\;\forall i\in N$$
(34)$$\varepsilon \leq P_{\theta_{i},\kappa_{i}}(r_{\theta_{i}},m),P_{\kappa_{i},\zeta }(r_{\kappa_{i}},m)\leq 1\;\;\forall i\in N,\;m\in M$$
(35)$$\varepsilon^{v+2}\leq Z_{\theta_{i},\kappa_{i}},Z_{\kappa_{i},\zeta }\leq 1\;\;\;\;\forall i\in N$$
(36)$$\varepsilon^{2}\leq x_{\theta_{i}}\leq C_{\theta_{i}}^{a}(r_{\theta_{i}},m),\varepsilon^{2}\leq x_{\kappa_{i}}\leq C_{\kappa_{i}}^{a}(r_{\kappa_{i}},m),\varepsilon^{2}\leq x_{\zeta }\leq C_{\zeta }^{a}(r_{\zeta },m)\;\;\;\;\forall i\in N,\;m\in M$$
(37)$$\varepsilon^{2}\leq y_{\theta_{i}}\leq C_{\theta_{i}}^{b}(r_{\theta_{i}},m),\varepsilon^{2}\leq y_{\kappa_{i}}\leq C_{\kappa_{i}}^{b}(r_{\kappa_{i}},m),\varepsilon^{2}\leq y_{\zeta }\leq C_{\zeta }^{b}(r_{\zeta },m)\;\;\;\;\forall i\in N,\;m\in M$$
(38)$$C_{\theta_{i}}^{\tau }(\min R_{\theta_{i}},m)\times \varepsilon \leq \beta_{\theta_{i},\kappa_{i}}\leq C_{\theta_{i}}^{\tau }(\max R_{\theta_{i}},m)\;\;\;\;\forall i\in N,\;m\in M$$
(39)$$C_{\kappa_{i}}^{\tau }(\min R_{\kappa_{i}},m)\times \varepsilon \leq \beta_{\kappa_{i},\zeta }\leq C_{\kappa_{i}}^{\tau }(\max R_{\kappa_{i}},m)\;\;\;\;\forall i\in N,\;m\in M$$
(40)$$\varepsilon^{n+v+4}\leq D_{\theta_{i},\zeta }\leq 1\;\;\;\;\forall i\in N$$
(41)$$r_{\theta_{i}}\in R_{\theta_{i}},r_{\kappa_{i}}\in R_{\kappa_{i}},r_{\zeta }\in R_{\zeta }\;\;\;\forall i\in N$$

### 4.3. Solution of LR Subproblems

After the LR problem was developed, it was decomposed into straightforward solvable subproblems according to different decision variables. We take the logarithm of the decision variables that were initially multiplied to make the multiplication into addition and bring the value of the range of each feasible solution into each subproblem. For example, the feasible solution of the decision variable is a continuous range between 1 and *ε*. This study differentiates its original formula once to obtain its extreme value and inserts the value of the first derivative and the two endpoints into the original subproblem to find the minimum value. On the basis of the range and of the concavity or convexity, each subproblem was easily solved in a few steps using algorithms or heuristics. A comprehensive set of LBs were determined to evaluate solution quality (GAP) for the primal and LR problems. Heuristic methods were designed to tune decision variables to satisfy all primal constraints and obtain a feasible solution. The proofs of subproblems are not listed in this study due to word count considerations.

### 4.4. Obtaining Primal Feasible Solution

The primal feasible solution is an algorithm developed by the study to find the optimal solution to the proposed problem (minimizing energy consumption). Each iteration finds a feasible solution closer to optimality, and this procedure continues until an optimal solution is reached. The results of the primal feasible solution can be compared with the dual problem (LR results) to judge the algorithm’s quality.

After solving each subproblem, we obtained a set of decision variables. We subsequently checked whether the decision variables were feasible. If they were, an UB was formed by the objective value of the primal problem. However, if the decision variables were not feasible, the use of heuristic methods was necessary to tune the decision variables into feasible solutions. In this study, we proposed obtaining the primal feasible solution for the tree structure sensor network based on its characteristics.

This model considers delay constraint (2) and throughput constraint (5) to obtain the primal feasible solution. We assume that the relay nodeκhas two subsystems that receive $q_{\kappa R_{i}}$ and send $q_{\kappa S_{i}}$, respectively. 

When $q_{\theta_{i}}=1/v$, $q_{\kappa S_{i}}=1/n$ and $q_{\kappa R_{i}}=1,\;P_{\theta_{i},\kappa_{i}}(r_{\theta_{i}},m)=1,\;q_{\zeta }=1,\;P_{\kappa_{i},\zeta }(r_{\kappa_{i}},m)=1$. The time slot required in delay constraint (2) is minimized. When throughput constraint (5) is violated, the value of the decision variable can be set as 1 to obtain the maximum throughput value. After adjusting the decision variables, we gradually decreased $q_{\kappa S_{i}}$, $q_{\zeta }$ and $r_{\kappa_{i}}$ to optimize the objective value for the primal problem and maintain the feasibility of the solution. The flow chart of the primal feasible solution is shown in Figure 4.

## 5. Experimental Results and Discussion

WSN experiments were conducted to verify the validity and performance of our proposed method. This study aimed to minimize the energy consumption of the tree structure WSN. 

A and B were the function probability functions used to transmit a packet with a *m* ∈ *M* transmission range radius of *r_i_* ∈ *R_i_*. When we set the transmission range radius *r_i_* to fixed, the higher the packet size m is, the lower the probability of transmitting a packet is. However, when we set the packet size m to fixed, the higher the transmission range radius *r_i_* is, the stronger the signal, the higher the signal-to-noise ratio (SNR) rate, and, therefore, the higher the probability of transmitting a packet is. The probability would be concave, which will be saturated or asymptotically converge to 1.

In order to implement the transmission probability, this study used the energy consumption characteristics of MICA2 motes [52] equipped with CC1000s [53]. Mica2 motes are the most commonly used sensor nodes in experimental WSN research due to their favorable energy dissipation properties [56], and CC1000 is a small electronic device used to transmit and receive radio signals between two devices, and a data cited from the above literature on the different distance between different sensor nodes were tested. Figure 5 shows the probability of successful transmission in three different SNR noise powers (power levels) based on the model developed by this study and the transmission data will be applied in the following experiments. 

### 5.1. Many-to-One WSN Transmission Experiment with Different Delay Scenarios

This study first conducted a data aggregation experiment with transmission from multiple nodes to relay nodes in a WSN environment in a many-to-one mode. We set the delay constraint *T_ik_* to 249.37585529, Figure 6a, in an experiment to test the performance of our proposed method. The distance between nodes was 30 m. The timeout interval *τ_i_* was set to 10 time slots. Ten sensor nodes were used; the results are shown in Figure 6a,b and Table 3. The blue line represents the Upper Bound (Primal Feasible Solution, the proposed solution in this study) and the orange line represents the Lower Bound (LR Solution). The unit of the horizontal axis is the number of iterations, and the unit of the vertical axis is the objective value as watts. This study also conducted an experiment using five sensor nodes and a tight delay constraint *T_ik_* of 113.1925051, Figure 6b. 

The results shows that no matter whether the delay of the system is set normal (249.37585529) or tight (113.1925051), or there is a different number of sensor nodes, the energy consumption is similar at 71.9 and 73.3 watts, respectively; the gap between LB and UB was 4.08 and 2.04, respectively. However, the decision time will be slightly slower when the number of nodes is larger at 15.8 s when n = 10 and 4.8 s when n = 5.

The probability of the sensor node being active when n = 5 was twice that of when n = 10. This is because when the delay constraint is tight, to obtain an optimal solution, the probability of a sensor node is close to 1/*n*. The probability that the relay node is active and transmitting a packet without error is close to 1. Therefore, total power consumption when n = 5 is similar to that of when n = 10 because there are half the number of nodes but the probability of each node being activated is twice as high. These results also show that the algorithm proposed in this study effectively and quickly achieved optimal results using different numbers of sensing nodes and delay values.

### 5.2. Tree Structure WSN Experiment with Different Numbers of Subtrees

A subsequent experiment was conducted to investigate the performance of the tree structure WSN model. The distance between nodes was 30 m. Each subtree had two sensor nodes (v), and the number of subtrees (n) was five, Figure 7a. The timeout interval *τ_i_* was set to 10 time slots. *T_θi,ζ_* was set to a tight value of 1001. Results are shown in Figure 7a,b and Table 4. The blue line represents the Upper Bound (Primal Feasible Solution, the proposed solution in this study) and the orange line represents the Lower Bound (LR Solution). The unit of the horizontal axis is the number of iterations, and the unit of the vertical axis is the objective value as watts. The experiment was also conducted using three sensor nodes in each subtree (v) and eight subtrees (n) for a bigger scale, Figure 7b, and *T_θi,ζ_* was set to 3000.

$T_{\theta_{i},\zeta }$$T_{\theta_{i},\zeta }$ The results show that whether the throughput of the system is set normal (1001) or tight (3000), or the different tree structure sensor networks, the power consumption is 332.75 and 498.06, respectively; the execution time was 308.6 and 4662.7, respectively, which is in the very tight allowable end to end delay (*T_θi,ζ_*). The gap between LB and UB was still under 5% which means the method is truly near-optimal. As the complexity of the tree structure increases, its energy consumption also increases to 332.75 watts and 498.06 watts, respectively.

The model adjusts the activation probability of the sink and relay nodes as much as possible to satisfy the delay and throughput constraints. If the throughput constraint remains unsatisfied, the probability is reduced according to the size of the multipliers to make it feasible. When solutions are feasible, decision variables are adjusted slightly to search for an optimal solution. Although more computing time can be expected for more complex tree structures, the algorithms proposed in this study all obtained energy-optimized solutions with smaller GAP.

### 5.3. Performance Analysis

This study’s aim of reducing WSN energy consumption is affected by the probability of each node being active, thereby reducing power consumption and increasing transmission efficiency. The advantage of an exhaustive search is its ease of implementation. If a solution exists, an exhaustive search finds it. This method is also used when the simplicity of implementation is more critical than speed. Compared with other complex algorithms, the model in this study avoids the problem of a probable blind spot [57]. On the other hand, the Round Robin (RR) algorithm was also applied in the experience. RR is an algorithm that is often used in practice in distributed computing scheduling [58]. The shorter the time slot, the higher the efficiency, which is suitable for the sensor network’s short-term and fast data transmission. Its advantage is that it can ensure data were distributed in each node equally, similar to the concept of fairness in this study. Also, fast data transmission makes the overall system use quality better.

We compared the experimental performance and efficiency of our proposed method against exhaustive search and RR in a test where all possible solutions for *q_i_* and *q_κ_* from 0.01 to 0.99 were searched for. The results are shown in Table 5.

The method proposed in this study only took 19.6 s, which is considerably quicker than the 315.5 s and 30.2 s used in the exhaustive search and RR, respectively, to find the optimal solution. The experiments show that an exhaustive search takes a long time to optimize energy consumption. However, RR causes higher energy consumption which takes a shorter time but higher energy consumption than an exhaustive search. The method proposed is superior to the previous two in terms of time and energy consumption, which demonstrates the efficiency of this study.

## 6. Conclusions

### 6.1. Contributions and Key Results

This study proposed an optimization-based power control mechanism to minimize the energy consumption in green WSNs and satisfy delay and throughput constraints. The tree structure WSN model was implemented in various scenarios and on a large and small scale. The fairness requirements were specially expressed in certain constraints to meet the practical application environment. After modeling, we applied the LR method to simplify the original problem. We then conducted a series of experiments to verify that our proposed method performed better than other heuristics. For the tree structure WSN model, we assumed that the relay node comprised two subsystems to control the probability of activation when sending data. The connection between these two subsystems may involve other problems and additional energy consumption. We compared system performance when this assumption was made versus when the assumption that only a single system controls the probability of activation when receiving and sending data was made. We hypothesize that the model is more flexible when the relay node is separated into two subsystems. Adjustments for the throughput constraint are made more quickly to obtain the primal feasible solution in our approach than in competing approaches. 

The gaps between UP and LB were all lower than 5%, which represents that the proposed method is very high quality and is truly near-optimal. The number of iterations and processing time also shows that the algorithm does not require an overly high-end computing device to perform WSN-compliant features.

### 6.2. Future Work

This study has two suggested directions for future work. The first would be to focus on the expansion of mathematical models. For example, to develop a more flexible model with multiple power consumption functions. It is also a topic worthy of attention to add the related considerations of device computing performance to the sensors at all levels. On the other hand, it can also be extended from the model of this study. Future researchers could look for objects of industry cooperation to introduce this method into the operation of enterprises so that the setting of various parameters is closer to the actual application scenario.

## Figures and Tables

**Figure 1 sensors-23-00763-f001:**
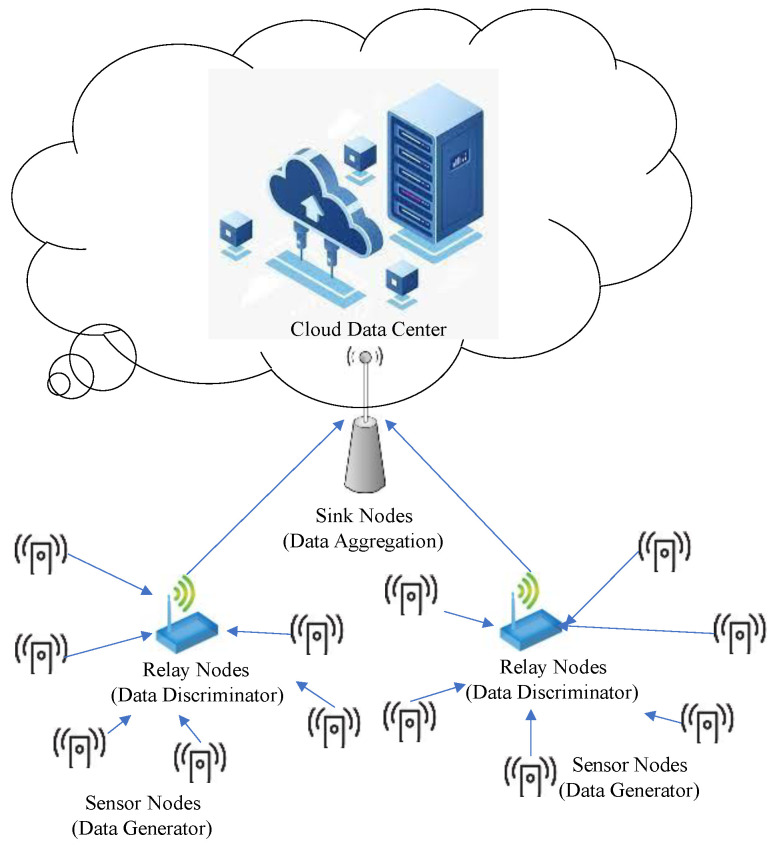
WSN Architecture with Data Aggregation in Different Levels of Sensor Nodes.

**Figure 2 sensors-23-00763-f002:**
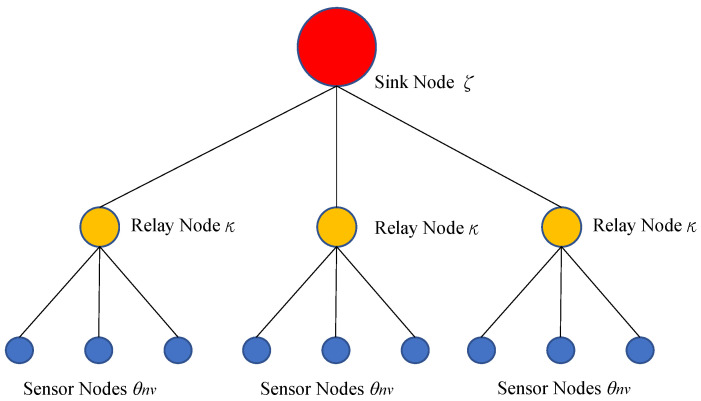
Tree Structure Model of WSN.

**Figure 3 sensors-23-00763-f003:**
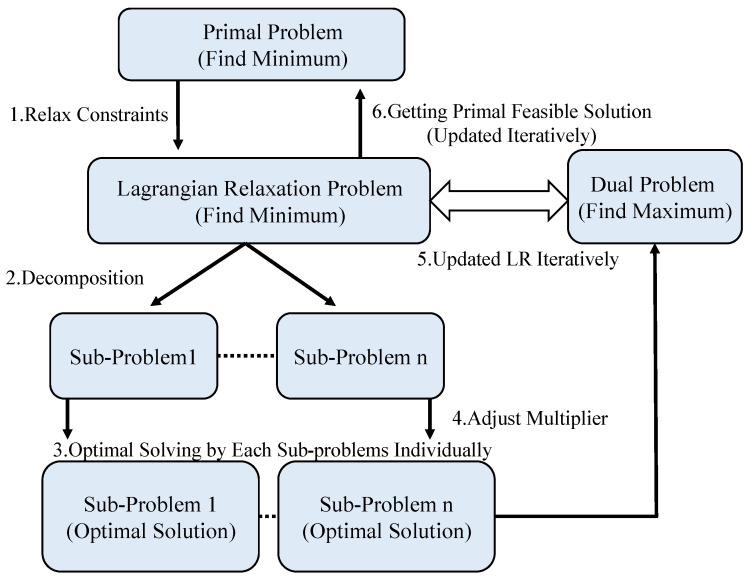
Procedure Flow Chart of Lagrangian Relaxation.

**Figure 4 sensors-23-00763-f004:**
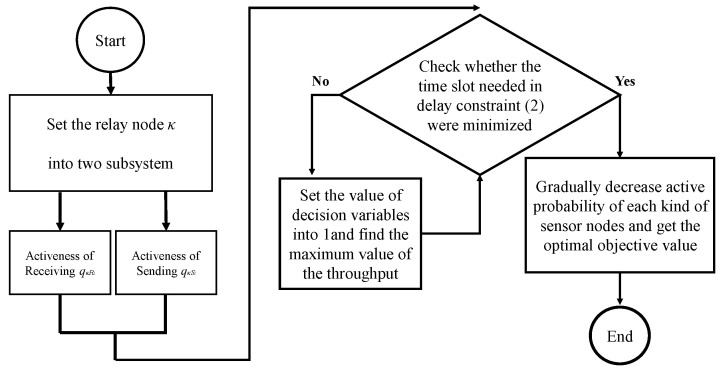
Follow Chart of the Proposed Primal Feasible Solution.

**Figure 5 sensors-23-00763-f005:**
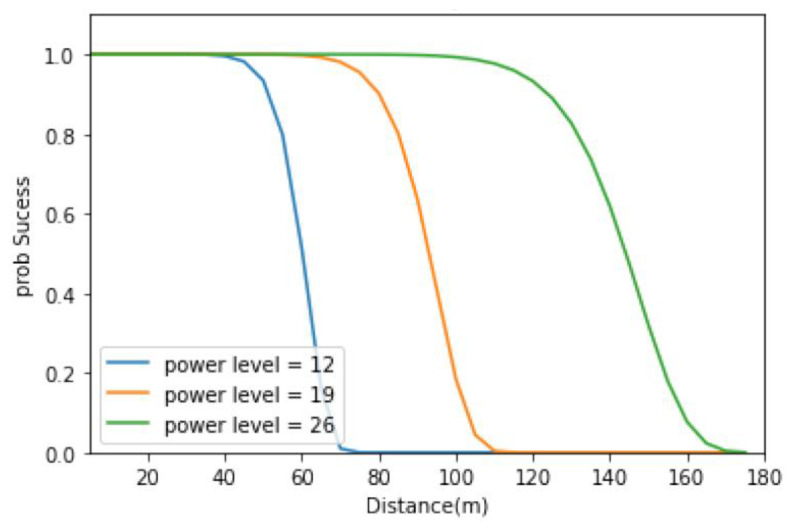
Probability Experience of Different Power Level and Distance Between Sensor Nodes.

**Figure 6 sensors-23-00763-f006:**
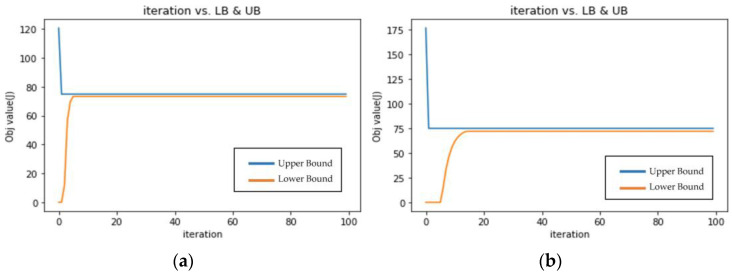
(**a**) Result when the delay constraint is set to 249.37585529 in many-to-one WSN transmission; and (**b**) result when the delay constraint is set to 113.1925051 in many-to-one WSN transmission.

**Figure 7 sensors-23-00763-f007:**
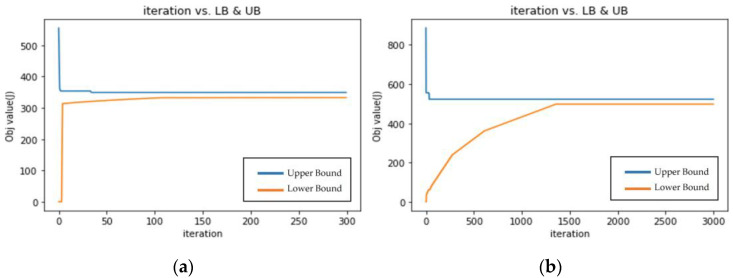
(**a**) Result when *T_θi,ζ_* is set to 1001; (**b**) Result when *T_θi,ζ_* is set to 3000.

**Table 1 sensors-23-00763-t001:** Given Parameters.

Notation	Description
N	The index set of possible number of subtrees, defined as $\left\{1,2,3,....\bar{n}\right\}$
V	The index set of possible numbers of sensor nodes in a single subtree, defined as $\left\{1,2,3,....\bar{v}\right\}$
$\theta_{ij}$	The sensor node that responsible for sensing and gathering data from different area, which is the node number j∈V in subtree i∈N
$\kappa_{i}$	The relay node from subtree i∈N, responsible for aggregating data sent from the lower-level layer sensor nodes
ζ	The sink node that responsible for aggregating data sent from the relay nodes
$\bar{T_{\theta_{i},\zeta }}$	The allowable end to end delay from sensor node to sink node for Subtree i∈N
$d_{\theta_{i},\kappa_{i}}$	Distance between sensor and relay node from subtree i∈N
$R_{\theta_{i}}$	Set of possible range for sensor nodes in subtree i∈N
$R_{\kappa }$	Set of possible range for relay node in subtree i∈N
$R_{\zeta }$	Set of possible range for sink node
$\tau_{\theta_{i}}$	Sensor node’s timeout interval from subtree i∈N (a given unit of time slots)
$\tau_{\kappa_{i}}$	Relay node’s timeout interval from subtree i∈N (a given unit of time slots)
$t_{\theta i,}\kappa_{i}$	The transmission time from sensor nodes to relay node, e.g., one time slot
t	The expected smallest time slots for the network to maintain
$P_{\theta_{i}}$	The initial power storage for sensor node i∈N
$P_{\kappa_{i}}$	The initial power storage for relay node from subtree i∈N
$P_{\zeta }$	The initial power storage for sink node ζ

**Table 2 sensors-23-00763-t002:** Decision Variables.

Notation	Description
$r_{\theta_{i}}$	The transmission range of sensor nodes in subtree i∈N, $r_{\theta_{i}}\in R_{\theta_{i}}$
$r_{\kappa_{i}}$	The transmission range of relay node in subtree i∈N, $r_{\kappa_{i}}\in R_{\kappa }$
$r_{\zeta }$	The transmission range of sink node
*M*	Index set of all possible packet size, including $\left\{1,2,3,....\bar{m}\right\}$
$q_{\theta_{i}}$	The active probability of sensor nodes in a time slot in subtree i∈N
$q_{\kappa R_{i}}$	The active probability of relay nodes receive data in a time slot in subtree i∈N
$q_{\kappa S_{i}}$	The active probability of relay nodes send data in a time slot in subtree i∈N
$q_{\zeta }$	The active probability of sink nodes in a time slot
$P_{\theta_{i},\kappa_{i}}(r_{\theta_{i}},m)$	The probability of sensor nodes to transmit packet with m∈M size to relay node in subtree i∈N when no error occurs with transmission range radius of $r_{\theta_{i}}\in R_{\theta_{i}}$
$P_{\kappa_{i},\zeta }(r_{\kappa_{i}},m)$	The probability of the relay node in subtree i∈N to transmit packet with m∈M M size to sink node when no error occurs with transmission range radius of $r_{\kappa_{i}}\in R_{\kappa }$
$C_{\theta_{i}}^{a}(r_{\theta_{i}},m)$	The average power consumption rate influenced by *m* when sensor nodes in subtree i∈N is active with transmission range $r_{\theta_{i}}\in R_{\theta_{i}}$ in one time slot
$C_{\theta_{i}}^{b}(r_{\theta_{i}},m)$	The average power consumption rate influenced by *m* when sensor nodes in subtree i∈N is inactive with transmission range $r_{\theta_{i}}\in R_{\theta_{i}}$ in one time slot
$C_{\kappa_{i}}^{a}(r_{\kappa_{i}},m)$	The average power consumption rate influenced by *m* when the relay node in subtree i∈N is active with transmission range $r_{\kappa_{i}}\in R_{\kappa }$ in one time slot
$C_{\kappa_{i}}^{b}(r_{\kappa_{i}},m)$	The average power consumption rate influenced by *m* when the relay node in subtree i∈N is inactive with transmission range $r_{\kappa_{i}}\in R_{\kappa }$ in one time slot
$C_{\zeta }^{a}(r_{\zeta },m)$	The average power consumption rate influenced by *m* when the sink node is active with transmission range $r_{\zeta }$ in one time slot
$C_{\zeta }^{b}(r_{\zeta },m)$	The average power consumption rate influenced by *m* when the sink node is inactive with transmission range $r_{\zeta }$ in one time slot
$C_{\theta_{i}}^{\tau }(r_{\theta_{i}},m)$	The average power consumption rate for sensor nodes in subtree i∈N to transmit a m∈M size packet in one time slot with transmission range $r_{\theta_{i}}\in R_{\theta_{i}}$
$C_{\kappa_{i}}^{\tau }(r_{\kappa_{i}},m)$	The average power consumption rate for the relay node in subtree i∈N to transmit a m∈M size packet in one time slot with transmission range $r_{\kappa_{i}}\in R_{\kappa }$

**Table 3 sensors-23-00763-t003:** LR Results of a Many-to-one WSN Transmission Experiment.

Experiments	LB (watts)	UB (watts)	Time (sec)	GAP (%)
Normal Delay	71.9	74.82	15.8	4.08
Tight Delay	73.3	74.84	4.8	2.04

**Table 4 sensors-23-00763-t004:** LR Results of Many-to-many WSN Transmission Experiment.

Experiments	LB (watts)	UB (watts)	Time (sec)	GAP (%)
$T_{\theta_{i},\zeta }$ = 1001	332.75	349.38	308.6	4.9
$T_{\theta_{i},\zeta }$ = 3000	498.06	522.76	4662.7	4.96

**Table 5 sensors-23-00763-t005:** Performance Analysis Results.

	Exhaustive Search	Round Robin (RR)	Proposed Method
Objective value (watts)	64.458	73.464	63.453
Time (sec)	315.5	30.2	19.6

## Data Availability

Not applicable.

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
