# Peer review of "A Near-Optimal Energy Management Mechanism Considering QoS and Fairness Requirements in Tree Structure Wireless Sensor Networks"

_sensors, 2023, doi:10.3390/s23020763_

Round 1

Reviewer 1 Report

This paper tackles the energy-efficient data transmission problem as a tree structure wireless sensor network mathematical model, which mainly considers the QoS and fairness requirements. This work determines the probability of sensor activity, transmission distance, and transmission of the packet size and thereby minimizes energy consumption by using the   Lagrangian Relaxation method. The numerical results show that the decision-making speed and energy consumption can be effectively improved.

On the other hand,  the paper has too many parameters, which is very difficult to follow. The number of parameters should be reduced and then the solution should be considered. Therefore, by also considering the following issues, the paper is not acceptable in its current form although the paper has a merit.

MAJOR ISSUES

+ Introduction section should be improved to give the motivation more clearly.

+ The main contributions of the paper should be clearly given as a separate subsection in the introduction section.

+ The organization of the paper should be clearly given as a separate subsection in the introduction section.

+ Especially considering the popularity of this problem, the related work should be improved. The number of references are insufficient. The related work and bibliography should be improved.

+ Most of the references in this paper are mostly recent publications (within the last 5 years) and relevant. On the other hand, the bibliography should be improved by adding most recent references.

+ Problem Definition and System Model are provided clearly as a separate section.

+ Preamble information between section "4. Optimization Solution Approach" and subsection "4.1 Lagrangian Relaxation Method" should be improved.

+ The proposed scheme performs well. The motivation behind it should be explained better.

+ In line 285 in Section 4.4, before "The time slot...", a few terms are separated by comma and the last one is equal to 1; however, it is not clear whether the previous three terms are also equal to 1 or not.

+ The figures/schemes are generally clear. They show the data properly. It is not difficult to interpret and understand them. On the other hand, Figure 1 should be explained better by adding more information to its caption and also give more details about some components like generator and discriminator in the figure.

+ Section "5. Experimental Results and Discussion" should be improved. Figures should be clearly explained, especially in the text/main body of the paper.

+ In the subfigures of Figure 5 and 6, the legends of the line should be given. There are two lines in each of those figures and what each line corresponds to cannot be understand. 

+ The conclusion should be improved by giving the key results and main contributions more clearly.

+ Future work part should be given in the conclusion section.

 MINOR ISSUES

+ The grammatical errors and typos should be fixed.

+ Size of Figure 3 should be increased.

+ Figure 5 and 6 should not exceed page margins so its size should be reduced.

+The odd places at the bottom of page 10 and page 13 should be filled with the following text content.

+ The references in the bibliography should be given in the same style. The following link should be checked: https://www.mdpi.com/authors/references 

Reviewer 2 Report

The work is interesting and tackles an up-to-date topic. Below are my six major concerns about this work:

(1) I think the authors should illustrate the problem defined with a detailed explanation. 

(2) Also, the authors need to explain more of the author's philosophy and the relation of the problem on QoS and fairness which are mentioned in the title. As it is explained that sensor nodes are various according to the priority and level of importance, and these characters are necessary to be reflected. 

(3)In 4.2 reformulation for LR Optimization solution, the logarithmic function was introduced, and in 4.3, the process of subproblem proof is omitted, more details with respect to these methods should be provided.

(4) In obtaining a feasible solution, the LR method is used and compared to the exhaustive search method. Although the exhaustive search is the worst case in terms of time, it can find the exact optimal value. Since the proposed technique is profitable in terms of time but shows near optimality, it is necessary to explain the justification for using the proposed technique in this respect. And performance comparison with other near-optimal techniques is also needed.

(5)It looks like the authors investigated a static scenario instead of a dynamic and time-varying one. Thus, the authors should better give some numerical simulations to verify that the proposed method can be applied in dynamic scenarios. Besides, the robustness of the proposed method should be discussed to show that this method is adapted to variable network topology and data saturation cases due to the heavy nodes.

(6) In order to show the superiority of the author's proposed method, performance M&S for QoS and fairness that can be obtained through the proposed method compared with the energy management mechanism for various WSNs is needed.

Reviewer 3 Report

In this paper, the authors proposed an optimization-based power control mechanism to minimize the energy consumption in green WSNs and satisfy delay and throughput constraints. Some suggestions are shown as follows.

1. The objective function should be explained more including the convex or non-convex, physical meaning and something related with implementation.

2. From Eq. 32 to Eq. 54, the authors are suggested to compact all these constraint together to better present their meaning.

3. In section 4.4, the authors mentioned "Obtaining a Primal Feasible Solution" Please explain more because the current presentation is not clear.

Round 2

Reviewer 1 Report

The paper is revised considerably based on my comments on the previous version of the paper. On the other hand, the paper should still be revised by considering the following issues:

MAJOR ISSUES

+ Introduction section should be improved to give the motivation more clearly.

+The author consider the energy-aware cluster-based routing problem with static sinks as it has been considered for two decades (since the Heinzelman' work published in 2000). On the other hand, researchers consider this problem with mobile sink to reduce the overall energy consumption of the network and increase it s lifetime for nearly one decade. The authors should present their motivation why they consider the problem with static sink. 

+The related work should include the following paper which tackles the energy aware routing problem with a mobile sink (a UAV with limited battery capacity). Thus, the authors can clarify not only their main contributions but also motivation of this paper in the related literature.

- O. M. Gul, A. M. Erkmen and B. Kantarci, "UAV-Driven Sustainable and Quality-Aware Data Collection in Robotic Wireless Sensor Networks," in IEEE Internet of Things Journal, vol. 9, no. 24, pp. 25150-25164, 15 Dec.15, 2022, doi: 10.1109/JIOT.2022.3195677.

+ Most of the references in this paper are mostly recent publications (within the last 5 years) and relevant. On the other hand, the bibliography should be improved by adding most recent references.

+ Preamble information between section "3. Mathematical Model" and subsection "3.1 Problem Definition" should be improved.

+ The figures/schemes are generally clear. They show the data properly. It is not difficult to interpret and understand them. On the other hand, Figure 1 should be explained better by adding more information to its caption and also give more details about some components like generator and discriminator in the figure.

+ Section "5. Experimental Results and Discussion" should be improved. Figures should be clearly explained, especially in the text/main body of the paper.

+ In the subfigures of Figure 5 and 6, the legends of the line should be given. There are two lines in each of those figures and what each line corresponds to cannot be understand. 

 MINOR ISSUES

+ The grammatical errors and typos should be fixed.

+ The references in the bibliography should be given in the same style. The following link should be checked: https://www.mdpi.com/authors/references 

Reviewer 2 Report

Ok. My questions are completed.

Author Response

We sincerely thank the reviewer again for conducting the review and giving us the valuable comments and feedback.

Reviewer 3 Report

The authors have revised the manuscript based on the reviewers' comments. I suggest to accept this paper.

Author Response

(The authors gave the same response as above.)
